# Development of Quebracho (*Schinopsis balansae*) Tannin-Based Thermoset Resins

**DOI:** 10.3390/polym13244412

**Published:** 2021-12-16

**Authors:** Emanuele Cesprini, Primož Šket, Valerio Causin, Michela Zanetti, Gianluca Tondi

**Affiliations:** 1Land Environment Agriculture & Forestry Department, University of Padua, Viale dell’Università 16, 35020 Legnaro, Italy; emanuele.cesprini@phd.unipd.it (E.C.); michela.zanetti@unipd.it (M.Z.); 2Slovenian NMR Centre, National Institute of Chemistry, Hajdrihova 19, SI-1000 Ljubljana, Slovenia; primoz.sket@ki.si; 3Department of Chemical Sciences, University of Padua, Via Marzolo 1, 35131 Padova, Italy; valerio.causin@unipd.it

**Keywords:** flavonoid, resin, bio-sourced, bioplastic, wood adhesives, polymerization

## Abstract

One of the major challenges currently in the field of material science is finding natural alternatives to the high-performing plastics developed in the last century. Consumers trust synthetic products for their excellent properties, but they are becoming aware of their impact on the planet. One of the most attractive precursors for natural polymers is tannin extracts and in particular condensed tannins. Quebracho (*Schinopsis balansae*) extract is one of the few industrially available flavonoids and can be exploited as a building block for thermoset resins due to its phenol-like reactivity. The aim of this study was to systematically investigate different hardeners and evaluate the water resistance, thermal behavior, and chemical structure of the quebracho tannin-based polymers in order to understand their suitability as adhesives. It was observed that around 80% of the extract is resistant to leaching when 5% of formaldehyde or hexamine or 10% of glyoxal or furfural are added. Additionally, furfuryl alcohol guarantees high leaching resistance, but only at higher proportions (20%). The quebracho-based formulations showed specific thermal behavior during hardening and higher degradation resistance than the extract. Finally, these polymers undergo similar chemistry to those of mimosa, with exclusive reactivity of the A-ring of the flavonoid.

## 1. Introduction

In a context of growing interest in sustainable products and a circular economy, the need for high-performing bio-based solutions is rising exponentially [1,2]. In particular, bioplastics are an attractive alternative, because they have the potential to combine the high performance we are used to having with a lower environmental impact compared to oil-based synthetic resins [3,4].

Among the main natural resources of particular interest as building blocks for thermosetting resins, condensed tannins are one of the most attractive alternatives due to their availability and their phenol-like chemistry [5,6]. These polyphenols are industrially extracted from mimosa (*Acacia mearnsii*) and quebracho (*Schinopsis balansae*) for several applications, especially in the leather tannery row [7,8], in oenology [9,10], as antitumor and anti-oncogenic activities in pharmaceutical and medical applications [11,12], for thermal valorization [13,14], and water remediation [15,16]. As previously mentioned, these extracts were proved to crosslink with several hardeners, exploiting their phenol-like chemistry. In the study of Pizzi et al., as well as in that of Yurtsever and Sengil, stable polymers were formed by the reaction between quebracho and formaldehyde [17,18]. This flavonoid substrate was also used for the preparation of wood preservatives and adhesives for bio-based composites in combination with hexamine [19,20,21]. Quebracho’s chemistry was also exploited for the preparation of insulating foams, where furfuryl alcohol was applied as comonomer [22,23]. Another attractive reaction of Quebracho flavonoids occurs with dimethyl carbonate and hexamine, where Thébault et al. were able to produce urethane without the use of isocyanate [24,25]. Further successful exploitation of quebracho was the blending with commercial phenol–formaldehyde (PF) resins [26]. Here, the role of the condensed tannin was useful not only in decreasing the synthetic part but also in enhancing the physical properties and containing the formaldehyde emissions of the resulting resin [27]. Despite the number of studies performed, a systematic assessment of the polymerization parameters has not yet been presented [28].

In our work, industrially available quebracho extracts are proposed as building blocks for thermoset resins. Several curing agents were selected: formaldehyde, hexamine, glyoxal, maleic anhydride, furfural, and furfuryl alcohol. The effects of pH and hardener concentration in the curing process were investigated, and the resulting polymers were characterized for their leaching resistance, thermomechanical (TMA), and thermal (TGA) behavior as well as for their chemical structure (FT-ATR and solid-state ^13^C-NMR). Therefore, this research provides an overview of the properties of quebracho tannin-based sustainable polymers and the most suitable conditions for their synthesis.

## 2. Experimental

### 2.1. Chemicals and Reagents

Quebracho tannin extract (Fintan 737B) was kindly provided by the company Silvateam (S. Michele Mondovì, Cuneo, Italy), while furfural and furfuryl alcohol were provided by International Furan Chemical IFC (Rotterdam, The Netherlands). Formaldehyde and glyoxal water solutions (37% and 40%, respectively), hexamethylenetetramine (hexamine), maleic anhydride, sulfuric acid (98%), and sodium hydroxide were purchased from Alfa Aesar (Thermo Fisher, Waltham, MA, USA).

### 2.2. Methods

#### 2.2.1. Adhesive Preparation and Hardening

Quebracho tannin water solutions were prepared at 50% by weight under vigorous stirring. The pH of the homogeneous solution (pH = 6.7) was modified to 2, 4, 6, and 8 by adding 33% wt. solutions of sulfuric acid and sodium hydroxide. Subsequently, 5 g of the tannin solution was introduced into plastic test tubes and added to 1, 2.5, 5, 10, or 15% of hardener based on solid tannin. The test tubes were then tightly screwed and exposed to 100 ± 5 °C for 24 h to cure. The solid obtained was removed and ground with a mortar to obtain a fine powder that was stabilized over 24 h at room temperature. Table 1 summarizes the experimental design of this study.

#### 2.2.2. Leaching Test

The test consisted of keeping 1 g of dried tannin polymer powder in 50 mL of deionized water under magnetic stirring for 1 h. The solution was then filtered (paper filter 125 µm), and the leaching resistance was calculated by weighing the dried filtered material according to the following formula:(1)$$Leaching\;Resistance\;\left(\%\right)=\frac{\mathrm{Weight}\;\mathrm{after}\;\mathrm{leaching}\;}{\mathrm{Weight}\;\mathrm{before}\;\mathrm{leaching}}\times 100$$

#### 2.2.3. Thermal Mechanical Analysis (TMA)

Thermomechanical analysis was carried out with a TMA/SDTA840 Mettler Toledo (Mettler Toledo, Columbus, OH, USA) instrument equipped with a three-point bending probe. The samples were prepared by applying about 20 mg of tannin-based formulations (before curing) between two beech wood plies (15 mm × 5 mm × 1.5 mm). A non-isothermal method was applied: a 10 °C/min heating rate was set, and a cycle of 0.1/0.5 N force was applied on the specimens, with each force cycle lasting 12 s (6 s/6 s).

Through the relationship between force and deflection, Young’s modulus *MOE* was calculated for each tested case. The equation used to calculate the modulus of elasticity is shown below:(2)$$MOE=\left[\frac{L^{3}}{\left(4bh^{3}\right)}\right]\left[\frac{\Delta F}{\Delta f}\right],$$
where *L* is length, *b* the width, and *h* the height of the samples. *F* is the force applied, and *f* the deflection due to the applied load.

#### 2.2.4. Thermogravimetric Analysis (TGA)

An SDT 2960 Simultaneous DSC-TGA TA instrument (Waters—TA instruments, New Castle, DE, USA) was used to measure the thermogram of cured samples, applying a 10 °C/min heating rate from 30 °C up to 800 °C. The weight loss of the samples was recorded as a function of temperature. The tests were carried out under an N_2_-flow inert environment.

#### 2.2.5. ^13^C-NMR Spectroscopy

Solid-state NMR experiments of the tannin-cured and leached polymers (with 15% hardener) were performed on a Bruker AVANCE NEO 400 MHz NMR spectrometer (Bruker, Billerica, MA, USA) using a 4 mm CP-MAS probe. The sample spinning frequency was 15 kHz. The ^1^H-^13^C CP-MAS NMR experiments consisted of excitation of protons with a p/2 pulse of 3.5 ms, CP block of 2 ms, and signal acquisition with high-power proton decoupling. A total of ca. 2000 to 15,000 scans were accumulated with a repetition delay of 5 s. The chemical shifts were referenced externally using adamantine.

#### 2.2.6. ATR FT-IR Spectroscopy

The same samples analyzed by ^13^C-NMR were analyzed with a Thermo Fisher Nicolet NEXUS B70 (Thermo Fisher Scientifics, Waltham, MA, USA) FT-IR instrument equipped with an ATR accessory with a diamond crystal. The spectra were acquired with 32 scans from 4500 to 600 cm^−1^, and the region between 1800 to 600 cm^−1^ was reported and discussed.

#### 2.2.7. Data Analysis

The data obtained from the TMA, TGA analysis and the FT-ATR spectra were elaborated using OriginPro 8.5.0 SR1 software (OriginLab Corp., Northampton, MA, USA).

## 3. Results and Discussion

The hardening tests performed adding different crosslinkers have shown that it is possible to produce several polymers of quebracho. The only crosslinker that gave rise to limited curing was maleic anhydride, and it will not be further treated in this article. According to the reaction mechanism proposed by Tondi [29], maleic anhydride crosslinks mimosa tannin through its -OH in position C3. This suggests that this position for quebracho is less frequently hydroxylated or less sterically accessible due to the higher branching and the pyrogallic B-ring.

The solid adducts obtained by curing the quebracho extract formulations with formaldehyde, hexamine, furfural, and furfuryl alcohol at various pH underwent a leaching procedure, and the leaching resistances are reported in Figure 1.

Most of the solid polymers tested registered a leaching resistance >70% at different pHs. Formaldehyde, glyoxal, and furfural produced water-resistant polymers at every pH; however, formaldehyde reached 80 ± 1% at pH 4; glyoxal reached 76 ± 1% at pH 2; and furfural reached 84 ± 1% at pH 8. Hexamine and furfuryl alcohol cured the quebracho extract at high (6, 8) and low (2, 4) pHs, respectively. These findings are in line with those observed for the mimosa tannin extract polymers, because the leaching resistances observed were also higher than 70%, but the major difference was the more suitable pH for the different crosslinkers [29].

Considering the most successful pH for every hardener, the leaching resistance was monitored by adding different amounts of hardeners. In Figure 2, the leaching resistance trend is reported for the five crosslinkers.

The graph generally depicts an increase in leaching resistance by increasing the amount of hardener; however, not all the crosslinkers present the same behavior. Formaldehyde, glyoxal, and hexamine reached a plateau of leaching resistance at around 5% of crosslinker, which suggests that, with 5% of hardener, the activation is complete; the addition of further hardener does not affect the leaching resistance, and around 20% of extract will be leached out. Conversely, furfural and furfuryl alcohol increase their leaching resistance, constantly reaching values over 85% when 30% of hardener is added. This behavior is due to the capacity of furanic monomers to self-polymerize [30,31], involving a structural tightening that also keeps the less reactive fractions of the tannin extract embedded in the network. However, it was observed that, with 10% hardener, furfural reached over 80% leaching resistance, while furfuryl alcohol did not even reach 70%, lowering its applicative interest.

In order to understand and compare the reactivity of the different hardeners, a concentration of 15% was selected to monitor the behavior of the formulations as a function of the temperature through TMA and TGA.

In Figure 3, the thermomechanical behavior of the quebracho–hardener formulations is reported.

The TMA confirms that the temperature increase involves an increase in stiffness, meaning that the formulations cure. Most of the measured formulations start their hardening at around 80 °C, and they reach maximum curing at 160 °C.

MOE is defined as the ratio of strain and deflection. The TMA instrument applies constant strain during the temperature rise and registers the deflections; therefore, a lower deflection implies an increase in rigidity (MOE), to be attributed to the curing process, which tightens the polymer network. The highest value was reached by hexamine (2700 MPa), suggesting higher structural tightening and hence a higher crosslinking degree. This stiffness is reached in only 50 °C (from 100 to 150 °C), highlighting the outstanding curing rate (high slope) for hexamine to combine with the quebracho tannin formulations.

A similar curing rate was observed for glyoxal, which also reached a high rigidity (MOE = 2300 MPa). Glyoxal started the curing slightly later (at around 110 °C) and showed faster joint degradation shortly after reaching the maximum at around 130 °C. This behavior was also observed by Navarrete et al. [32]. This means that glyoxal starts curing later, increasing the stiffness similarly to hexamine, but the produced polymer suffers higher temperatures more than that with hexamine.

As expected from the leaching resistance tests, furfural increases the rigidity of the adduct stepwise (gentler slope), reaching high values of MOE (2490 MPa). Here, the curing starts at around 80 °C and increases until 160 °C, with a variable slope, accelerating at around 105 °C. The higher MOE is in line with the higher leaching resistance of this formulation (at 15% hardener).

Surprisingly, the formulation with formaldehyde registered relatively contained MOE (1980 MPa), which indicates a more elastic behavior when the polymer is cured. This could be explained by the evaporation of the excess of hardener, which also explains the earlier start at 80 °C.

Furfuryl alcohol recorded the lowest value of MOE, reaching only 1620 MPa, suggesting incomplete polymerization. Additionally, this assumption fits well with the leaching resistance experiments, where a higher amount of hardener would be needed (30%) to completely cure the formulation. Further, the curing occurs in two separate steps (110 °C and 130 °C), meaning that this hardener reacts with itself and with the flavonoids at different stages.

Table 2 highlights how the amount of hexamine affects the curing process of the formulations, reporting the initial curing temperature, the curing rate, and the maximum MOE registered in TMA experiments.

Firstly, it is possible to observe that initial curing temperature and amount of hardener are proportional. This seems to be due to the increase in viscosity of tannin solutions when hexamine is added, which is in line with the study of Moubarik et al. [33] that reported increasing viscosity for hexamine-added tannin solutions. The curing rate is expressed as the slope of the central part of the sigmoid of the thermogram, and it shows that higher amounts of hardener involve faster curing of the resin, although this kinetic does not affect the water resistance of the polymer. Finally, it has to be considered that the amount of hardener does not significantly affect the maximum elastic modulus, meaning that the final rigidity of the cured polymer is almost independent of the amount of hexamine added.

Once the polymers are synthesized, it was possible to observe their degradation when the temperature increased up to 800 °C in order to evaluate the behavior of the polymer during possible processing at high temperature. In Figure 4, the TGA of the five polymers is presented:

Thermogravimetric analysis provides information about the degradation due to increasing heat. Three main regions could be identified during the rise of temperature for all samples, and the weight loss percentage is reported in Figure 4. The first one (25–130 °C) characterizes mainly loss of water absorbed and traces of volatile compounds such as CO and CO_2_. The second region (135–600 °C) is typical of the degradation of later chains of the tannin and decomposition of hardeners, with the cleavage of C-C bonds forming CH_4_, CO, or CO_2_ [34,35]. In particular, two main events occur for a tannin degradation: the first, at around 260 °C, corresponds to decarboxylation, and the other one at, around 600 °C, corresponds to oxidation of high-carbon residues [36]. The last region between 600 and 800 °C is characterized by the decomposition of natural structures of tannins, in particular the disruption of the polyphenolic structure (rings A and B) [35]. Overall, it can be observed that the thermal degradation of the quebracho cured polymers is similar for all hardeners and presents a constant degradation pattern involving an acceleration between 250 and 300 °C for all polymers. This is different from what was observed for the quebracho powder, which degrades faster starting from 150–200 °C. This might suggest that the networking established after curing facilitates the initial rearrangement of the aromatic molecules during pyrolysis [37], involving the need for more energy before starting the linear degradation. Comparing the behavior of tannin resins with industrial resins (PF), this does not show any substantial differences in temperature resistance [38] but rather increases its resistance when added to an industrial resin (UF) [39]. This is a further detail that suggests the establishment of a real polymer network of the quebracho–hardener adducts.

In order to shed further light on the polymerization process, two chemical investigations of the cured polymers after leaching were also considered: ^13^C-NMR (Figure 5) and FT-IR.

Following the findings of the previous study of Tondi [29] with the polymers of mimosa tannin extracts, it is possible to observe that the major difference between quebracho and mimosa extract is the inverted area of the signals at 105 and 120 ppm.

The quebracho extract presents major absorption at around 120 ppm, while the mimosa extract showed higher absorption at 105 ppm. These two absorptions are related to the –C8 position of A and B (C2′, C5′, and C6′) rings, respectively, suggesting that quebracho has lower -OH groups in the A-ring (resorcinol-like) and/or more substituted (branched), while the B-ring also has lower OH groups, suggesting a catechol-like ring [40]. Figure 6 shows the dominant flavonoid structure of quebracho extract.

The fact that quebracho extract has lower signal intensity at around 105 ppm complicates the evaluation of further crosslinking; however, it can be observed that this band significantly decreases for all polymers except for the furfuryl alcohol (here, the CH of furans overlap). This means that the number of free C8 positions in the A-rings decrease, and hence, they are substituted by the activating hardeners when cured. Conversely, despite the higher number of free C in the catechol B-ring, they do not decrease for any hardener, suggesting that, in quebracho, activation and, therefore, polymerization through the B-ring do not occur. This observation is also in line with the chemistry of phenol: in case of the pyrogallic nature of the B-ring, the OH groups in positions 3′ and 5′ activate the 2′ and 6′ positions, and only the OH group in position 4′ is inhibitory; conversely, in quebracho, the catechol has vicinal OH groups in positions 3′ and 4′ that do not activate the positions 2′, 5′, and 6′.

Another observation to be highlighted is that the overall shape of the spectra of the cured polymers appears broader than that of quebracho extract, suggesting an increase in molecular dimension.

Finally, the ATR FT-IR spectrum of the leached polymer powders was collected, and it is reported in Figure 7.

A major observation can be made in the region between 1800 and 600 cm^−1^. The signal at 1550 cm^−1^ in the C=C aromatic stretching region disappears for every polymerized powder. This vibration is unique for quebracho and can be attributed to the C6 free position in the repeating unit connected C4–C8.

The band at 1510 cm^−1^ disappears for hexamine, and this occurs when the π electrons can be delocalized in three-dimensional networks, or alternatively, the band shifts to 1600 cm^−1^ into a bulky signal of aromatics where symmetric and asymmetric vibrations absorb at the same wavenumber.

The bands at 1400 cm^−1^ and at 1030 cm^−1^ decrease for most of the polymers, and they can be attributed to C-H bending, asymmetrical and symmetrical, respectively. These highlight an increase in steric hindrance that restrains some vibrations of the polymer. In this context, the disappearance of the signal at 970 cm^−1^ due to out-of-plane C-H bending is also logical. This signal remains only in the furfuryl alcohol polymer, strengthening the idea that some of the flavonoids are just “caged” into the furanic network.

## 4. Conclusions

Quebracho tannin extract can be successfully polymerized with five hardeners: formaldehyde, hexamine, glyoxal, furfural, and furfuryl alcohol. The leaching resistance of these polymers reaches values of around 80%, suggesting that part of the extract does not take part in the curing process and can be removed easily. The hardeners used cured with different kinetics: hexamine and glyoxal cured quickly in a one-step process, and furfural cured more stepwise. The produced polymers show enhanced thermal resistance compared to the quebracho extract, because the networking facilitates the rearrangement during pyrolysis. The polymerization of the formulations was also spectroscopically proven. Quebracho polymers are connected exclusively through the A-ring with every hardener, and the crosslinking products are similar to those occurring with the mimosa extract. Evidence of structure tightening can be seen in both spectroscopies, including band broadening and a decrease in C-H bending vibrations. The quebracho tannin-based formulations developed in this study can be proposed as alternative thermosetting polymers to replace phenolic resins in adhesives, coatings, insulation materials, and other molded products of the construction sector.

## Figures and Tables

**Figure 1 polymers-13-04412-f001:**
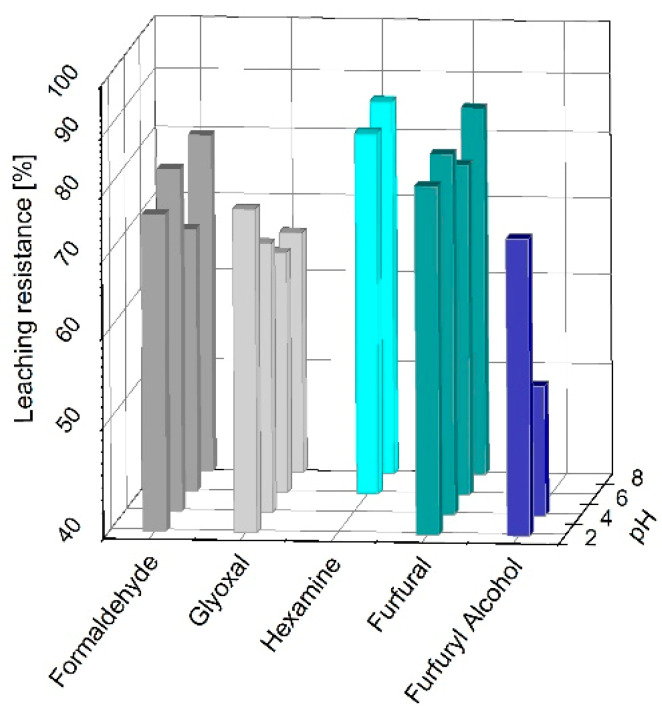
Leaching resistances of quebracho tannin formulations cured with 15% of formaldehyde, glyoxal, hexamine, furfural, and furfuryl alcohol at pH 2, 4, 6, and 8.

**Figure 2 polymers-13-04412-f002:**
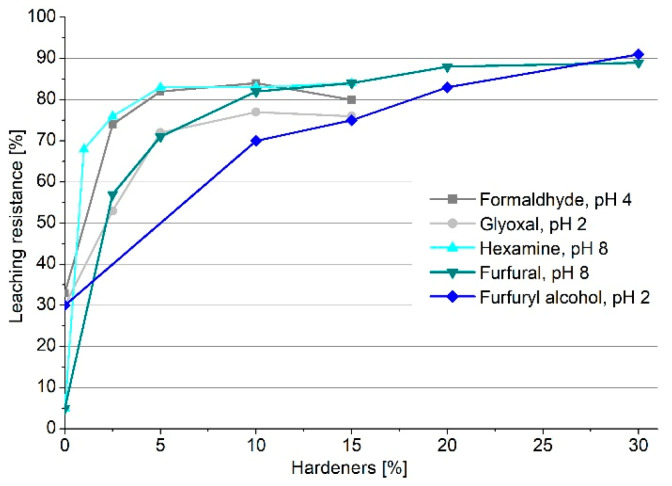
Leaching resistance of quebracho tannin polymers as a function of the amount of crosslinker.

**Figure 3 polymers-13-04412-f003:**
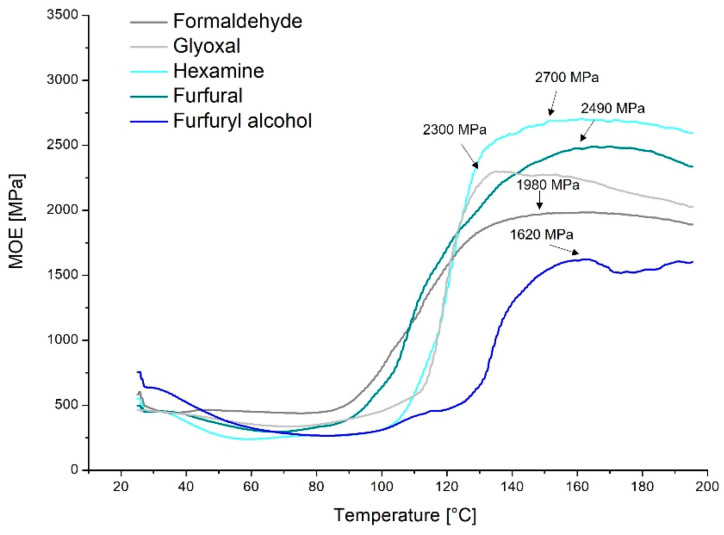
Thermomechanical analysis of the quebracho–crosslinker formulations.

**Figure 4 polymers-13-04412-f004:**
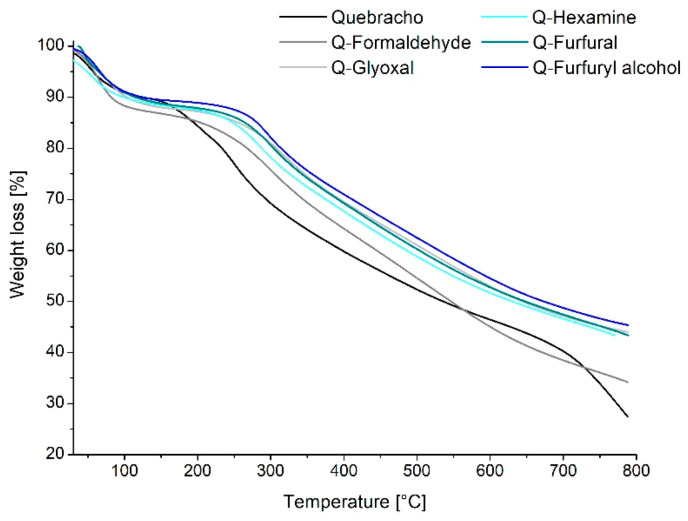
Thermogravimetric analysis of the quebracho-crosslinked polymers.

**Figure 5 polymers-13-04412-f005:**
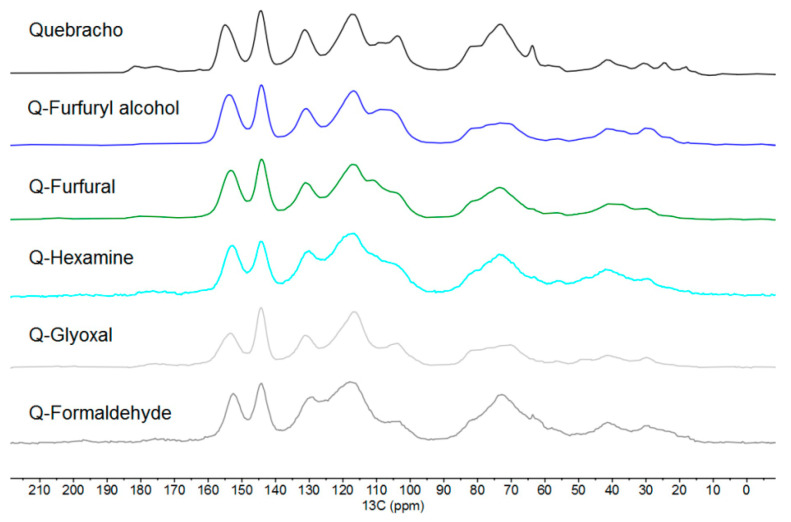
Solid-state ^13^C-NMR spectra of quebracho extract and its five polymers.

**Figure 6 polymers-13-04412-f006:**
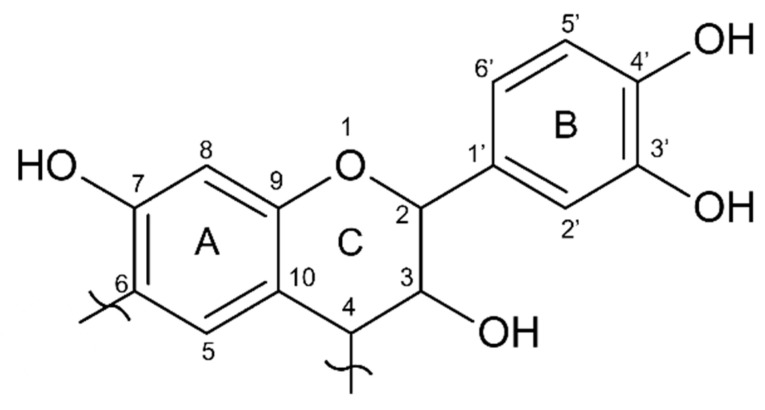
Model structure of the main flavonoid component of quebracho (Profisetinidin).

**Figure 7 polymers-13-04412-f007:**
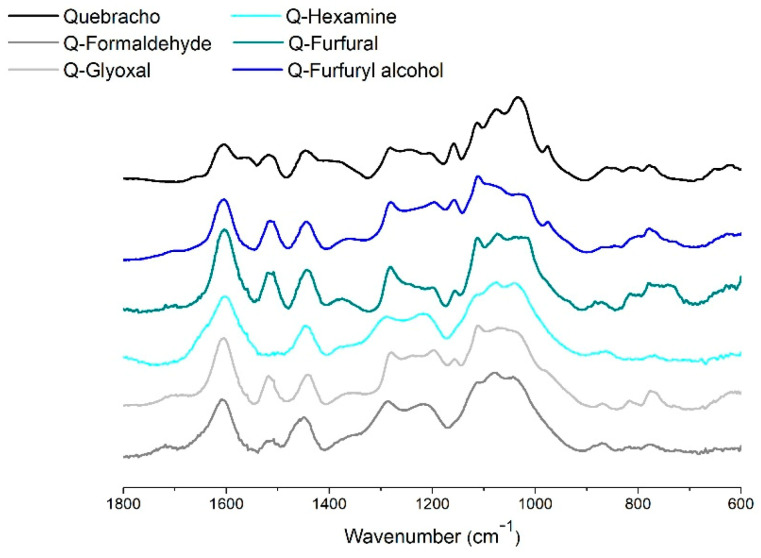
ATR FT-IR spectra of quebracho extract and its five polymers.

**Table 1 polymers-13-04412-t001:** Summary of the different hardeners studied by varying their concentrations and pH.

Hardeners	Amount	pH
Formaldehyde (37%)	2.5%, 5%, 10%, 15%	2, 4, 6, 8
Glyoxal (40%)	2.5%, 5%, 10%, 15%	2, 4, 6, 8
Hexamine (33%)	1%, 2.5%, 5%, 10%, 15%	2, 4, 6, 8
Furfural	2.5%, 5%, 10%, 15%, 30%	2, 4, 6, 8
Furfuryl alcohol	5%, 10%, 15%, 20%, 30%	2, 4, 6, 8
Maleic anhydride (50%)	5%, 10%, and 15%	2, 4, 6, 8

**Table 2 polymers-13-04412-t002:** Starting temperature (T_i_), curing rate, and maximum Young’s modulus (MOE max) for quebracho formulations at different amounts of hexamine.

Hexamine (%)	T_i_ (°C)	Curing Rate (MPa/°C)	MOE _max_ (MPa)
1	78	20 ± 0.3	2780
2.5	81	27 ± 0.3	2590
5	84	37 ± 0.5	2570
10	88	42 ± 0.5	2960
15	93	50 ± 0.7	2700

## Data Availability

The data of this study are guarded by authors.

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
