# Peer review of "Development of Quebracho (Schinopsis balansae) Tannin-Based Thermoset Resins"

_polymers, 2021, doi:10.3390/polym13244412_

Round 1

Reviewer 1 Report

Dear Editor,

dear Authors,

In the present work authors selected tannin extracts to develop and characterize thermoset resins toward an eco-friendly synthesis approach. The overall effort seems to be quite interesting. Some concerns are listed below:    

Title

The scientific name (Schinopsis balansae) of the Quebracho should be embedded in the title.

Introduction

Provide more citations to further strengthen your claims dealing with the literature data.

Experimental

-Give details for maleic anhydride used.

-Authors used the solid state 13C-NMR and ATR FTIR spectroscopy techniques to characterize the quebracho extract and its five polymers. They also measured the leaching resistance of the produced polymers. Why didn't you analyze the leached liquid material by means of an analytical technique (eg HPLC, GC-MS) in order to determine the amount of the individual compound? Give an explanation about this issue.

-Improve the font size of the Figure legends to be more visible.

Author Response

Dear Editor, dear authors,

First of all, we would like to thank you very much your engagement in reviewing our paper.

We have improved the manuscript according to the suggestions of the reviewers as much as possible; for the questions that were not processed we report here underneath our detailed answers.

In the text the new parts have been added in red.

REVIEWER 1

Dear Editor,

dear Authors,

In the present work authors selected tannin extracts to develop and characterize thermoset resins toward an eco-friendly synthesis approach. The overall effort seems to be quite interesting.

Some concerns are listed below:    

Title

The scientific name (Schinopsis balansae) of the Quebracho should be embedded in the title.

We agree with the reviewer, the scientific name was added in the title.

Introduction

Provide more citations to further strengthen your claims dealing with the literature data.

We agree with the reviewer, the literature was not complete. We have extended it in the text.

Experimental

-Give details for maleic anhydride used.

We are sorry for this omission, we have overworked the material part in a more systematic way.

-Authors used the solid state 13C-NMR and ATR FTIR spectroscopy techniques to characterize the quebracho extract and its five polymers. They also measured the leaching resistance of the produced polymers. Why didn't you analyze the leached liquid material by means of an analytical technique (eg HPLC, GC-MS) in order to determine the amount of the individual compound? Give an explanation about this issue.

Thanks for the commentary. We agree that this analysis would have given further information on the chemicals leached out. We suspect that the molecules solubilized were low molecular mass fractions of unreacted flavonoids and hydrocolloids fragments. This analysis could give us some more information on the composition of the quebracho extract, but we wonder if this could fit well in the description of the polymerization. At present we do not have chromatography facilities which would allow the detection of all components, but this remain a good idea for further investigations.

-Improve the font size of the Figure legends to be more visible.

We agree with the reviewer, we have slightly overworked every figure, to have more homogeneous graphics.

Reviewer 2 Report

The authors reported  Quebracho Tannin based materials to develop thermoset resins. The authors presented the importance of the natural alternatives to be used to replace chemicals. The authors organized and supported their finding in the way to prove their outcome. However, the manuscript still needs revision and there are things that need clarification. Therefore, please find below some comments/suggestions which might improve the quality of the manuscript:

  1. The authors claimed, “In our work, industrially-available quebracho extracts is proposed as building block for thermoset resins.” However, no description of any polymerization method was reported. Furthermore, the quantification of the modification should be mentioned.
  2. The authors stated “Despite the number of researches done, a systematic assessment of the polymerization parameters was not presented yet” but no reference was cited or supporting material. The authors are requested to provide relevant literature to support their statement.
  3. The authors perform leaching test; however, no standard method was mentioned for this test; the authors are requested to provide details on the standard test performed.
  4. The authors stated ” The only cross-linker which gave place to limited curing was maleic anhydride and it will not be further treated in this article” but no argument was inserted. To understand better this outcome, the authors are advised to provide additional details.

Author Response

Dear Editor, dear authors,

First of all, we would like to thank you very much your engagement in reviewing our paper.

We have improved the manuscript according to the suggestions of the reviewers as much as possible; for the questions that were not processed we report here underneath our detailed answers.

In the text the new parts have been added in red.

REVIEWER 2

The authors reported  Quebracho Tannin based materials to develop thermoset resins. The authors presented the importance of the natural alternatives to be used to replace chemicals. The authors organized and supported their finding in the way to prove their outcome. However, the manuscript still needs revision and there are things that need clarification. Therefore, please find below some comments/suggestions which might improve the quality of the manuscript:

  1. The authors claimed, “In our work, industrially-available quebracho extracts is proposed as building block for thermoset resins.” However, no description of any polymerization method was reported. Furthermore, the quantification of the modification should be mentioned.

Thanks for the commentary. We have performed this investigation based on previous study on mimosa [REF 29, 40]. The crosslinking of the mimosa flavonoid was reported in details in [29] and the quebracho flavonoid is not that different [40]. For this reason we have focused on the difference between the two.

  1. The authors stated “Despite the number of researches done, a systematic assessment of the polymerization parameters was not presented yet” but no reference was cited or supporting material. The authors are requested to provide relevant literature to support their statement.

We agree with the reviewer, the literature was not complete. We have extended it in the text.

  1. The authors perform leaching test; however, no standard method was mentioned for this test; the authors are requested to provide details on the standard test performed.

Thanks for the commentary. Similar tests are not easy to reproduce as standard because the leaching depends on the molecular mass of the polymer but also on their chemistry. To the best of our knowledge there are no standard methods to be considered because these test are not interesting for the adhesive industry – which is probably the closer to this subject.

  1. The authors stated ” The only cross-linker which gave place to limited curing was maleic anhydride and it will not be further treated in this article” but no argument was inserted. To understand better this outcome, the authors are advised to provide additional details.

We agree with the reviewer. We have extended it in the text the part related to maleic anhydride. Actually the polymer coming out was weaker and leachable to a high extent (ca.50%). For mimosa the results were far better and we have proposed in the text some reasons.

Round 2

Reviewer 2 Report

The Authors answered to the addressed queries.